# Visible-Light-Driven Antimicrobial Activity and Mechanism of Polydopamine-Reduced Graphene Oxide/BiVO_4_ Composite

**DOI:** 10.3390/ijms23147712

**Published:** 2022-07-12

**Authors:** Biyun Li, Xiaoxiao Gao, Jiangang Qu, Feng Xiong, Hongyun Xuan, Yan Jin, Huihua Yuan

**Affiliations:** School of Life Sciences, Nantong University, Nantong 226019, China; libiyun1986@163.com (B.L.); g13160495899@163.com (X.G.); qujiangang@ntu.edu.cn (J.Q.); xiongfengxl@163.com (F.X.); hyxuan_seu@163.com (H.X.)

**Keywords:** PDA-rGO/BiVO_4_, nanocomposite, antibacterial, mechanism

## Abstract

In this study, a photocatalytic antibacterial composite of polydopamine-reduced graphene oxide (PDA-rGO)/BiVO_4_ is prepared by a hydrothermal self-polymerization reduction method. Its morphology and physicochemical properties are characterized by scanning electron microscopy (SEM), energy-dispersive X-ray spectroscopy (EDX), Fourier-transform infrared (FT-IR), and X-ray diffraction (XRD). The results indicate that BiVO_4_ particles are evenly distributed on the rGO surface. *Escherichia coli* (*E. coli*) MG1655 is selected as the model bacteria, and its antibacterial performance is tested by flat colony counting and the MTT method under light irradiation. PDA-rGO/BiVO_4_ inhibits the growth of *E. coli* under both light and dark conditions, and light significantly enhances the bacteriostasis of PDA-rGO/BiVO_4_. A combination of BiVO_4_ with PDA-rGO is confirmed by the above characterization methods as improving the photothermal performance under visible light irradiation. The composite possesses enhanced photocatalytic antibacterial activity. Additionally, the photocatalytic antibacterial mechanism is investigated via the morphology changes in the SEM images of MG1655 bacteria, 2′,7′-dichlorofluorescein diacetate (DCFH-DA), the fluorescence detection of the reactive oxygen species (ROS), and gene expression. These results show that PDA-rGO/BiVO_4_ can produce more ROS and lead to bacterial death. Subsequently, the q-PCR results show that the transmembrane transport of bacteria is blocked and the respiratory chain is inhibited. This study may provide an important strategy for expanding the application of BiVO_4_ in biomedicine and studying the photocatalytic antibacterial mechanism.

## 1. Introduction

Currently, a variety of pathogenic bacteria seriously threaten human health, which is becoming a major concern for society [1]. There is a growing demand for effective and green environmental antibacterial materials. Photocatalytic materials have attracted widespread attention for the photocatalytic inactivation of bacteria in the field of biomedical applications [2].

BiVO_4_ is a kind of visible light photocatalyst that has the advantages of low bandgap energy, green environmental protection, low cost, high stability, and antibacterial activity [3,4]. However, there are some shortcomings of BiVO_4_, such as narrow band gap, poor adsorption capacity, and fast electron hole recombination rate [5]. In order to overcome the above issues, researchers have focused on the modification of BiVO_4_ with other materials [6,7,8,9,10,11,12]. Graphene-based nanomaterials, including graphene oxide (GO) and reduced graphene oxide (rGO), have been proved to have high antibacterial activity [13,14]. They have also been combined with BiVO_4_ to inhibit the recombination of electron-hole pairs and improve photocatalytic antibacterial activity by a synergetic effect [15,16,17,18,19]. El-Yazeed et al. [16] added GO to BiVO_4_, calcined it at 250 °C, and evaluated the antibacterial behavior against *Escherichia coli* (*E. coli*), *Bacillus subtilis*, and *Candida Albicans*. Pure BiVO_4_ showed little inhibitory activity (<50%) and the antibacterial activity of rGO/BiVO_4_ nanocomposites in the three strains was significantly improved. The synergy between the graphene sheets and BiVO_4_ particles plays an important role in enhancing antibacterial activity. However, the above synthesis methods for rGO/BiVO_4_ composites consume a lot of energy. Liao et al. [20] reduced GO with dopamine (DA) and then synthesized Ag-PDA-RGO nanocomposites in situ by adding AgNO_3_. The inhibition rates of GO, PDA-RGO, and Ag-PDA-RGO on *E. coli* were compared. Compared with GO and PDA-RGO, the antibacterial activity of Ag-PDA-RGO nanocomposites was significantly improved, and the antibacterial rate of *E. coli* reached 90.9%. Combining Ag with PDA-rGO significantly improved the antibacterial activity of nanocomposites. DA has recently been used as a green reducing agent for GO [21] and an adhesive agent for BiVO_4_ [22]. Nevertheless, BiVO_4_ modified with polydopamine-reduced graphene oxide (PDA-rGO) has rarely been reported to have antibacterial applications under visible light irradiation. Additionally, the antibacterial mechanism is not clear.

Hence, in this study, we prepare a photocatalyst of PDA-rGO /BiVO_4_ by a hydrothermal self-polymerization reduction method. The photocatalytic antibacterial activity of the synthesized composite is investigated against *E. coli* MG1655. The photocatalytic antibacterial mechanisms determined by bacterial morphological changes, the fluorescence detection of reactive oxygen species (ROS), and gene expression, are discussed.

## 2. Results and Discussion

### 2.1. PDA-rGO/BiVO_4_

The as-synthesized BiVO_4_ nanoparticles and PDA-rGO/BiVO_4_ composites were examined for morphology and purity by a scanning electron microscope (SEM) equipped with energy-dispersive X-ray spectroscopy (EDX) techniques, as shown in Figure 1A–D. For both pure BVO_4_ and the PDA-rGO/BiVO_4_ composites, the BVO_4_ particles exhibited well-fined flower-like 3D superstructures. The whole flower possessed a 3D eight-pot-shaped structure with high symmetry, which is consistent with previous reports [23,24]. EDX spectra showed the existence of Bi, V, O, and C elements, implying the purity of the samples, as shown in Figure 1C,D. Figure 1E shows the Fourier-transform infrared (FT-IR) spectra of BiVO_4_, PDA-rGO, and PDA-rGO/BiVO_4_. Both the PDA-rGO and the PDA-rGO/BiVO_4_ FT-IR spectra displayed vibration bands at 1720 cm^−1^, corresponding to C=O stretching vibrations [21,25]. The peaks at 720 cm^−1^ in both the BiVO_4_ and the PDA-rGO/BiVO_4_ composites corresponded to the asymmetric stretching vibration of metal-oxides due to the V-O group, and the shoulder peak at 673 cm^−1^ resulted from the shorter V-O bond, which might be due to the presence of both tetragonal and monoclinic phase structures. The reduction in the distinctive peak at 720 cm^−1^ clearly suggests the presence of a longer asymmetric V-O bond of the pure monoclinic phase [26,27], and these results are consistent with the morphology of BiVO_4_ and PDA-rGO/BiVO_4_ (Figure 1A,B). X-ray diffraction (XRD) patterns of the prepared BiVO_4_ nanoparticles and the PDA-rGO/BiVO_4_ composite are presented in Figure 1F. The diffraction pattern of BiVO_4_ and the PDA-rGO/BiVO_4_ nanocomposite clearly showed sharp peaks in the monoclinic scheelite -phase BiVO_4_ [24]. The presence of monoclinic scheelite -phase BiVO_4_ in the nanocomposite was confirmed by the (040) plane at 30.5° and the diffraction pattern was in accordance with the JCPDS card number (14-0688) of BiVO_4_ [24,26]. Furthermore, the BiVO_4_ peak intensity significantly increased with the addition of PDA-rGO to the composite of PDA-rGO/BiVO_4_. These results suggest that the crystalline nature of BiVO_4_ does not change due to the preparation of PDA-rGO and also clearly demonstrates that the BiVO_4_ nanosheets were successfully added into the PDA-rGO.

### 2.2. Photothermal Property

The photothermal effects of aqueous PDA-rGO/BiVO_4_ dispersions were evaluated by a digital infrared thermal imaging system (Figure 2). There was no significant difference in temperature between PDA-rGO/BiVO_4_ and BiVO_4_ before illumination (Figure 2B). As shown in Figure 2C, compared with pure BiVO_4_, PDA-rGO/BiVO_4_ showed a higher temperature when exposed to a xenon lamp for 10 min. Moreover, this trend was more obvious with the prolongation of light exposure time (Figure 2D). This effect is attributed to the good photothermal effect of PDA and graphene when stimulated by visible light [28,29].

### 2.3. Antibacterial Activity

The number of viable bacteria in the PDA-rGO/BiVO_4_ treatment group was significantly lower than that in the BiVO_4_ treatment group and the control group under light conditions (Figure 3). In addition, although the bacteriostasis effect was not obvious under dark conditions, the number of viable cells in the PDA-rGO/BiVO_4_ treatment group was also lower than that in the BiVO_4_ treatment group and the control group. These results indicate that PDA-rGO/BiVO_4_ can inhibit the growth of *E. coli* under both light and dark conditions, and that light significantly enhances the bacteriostasis of PDA-rGO/BiVO_4_. Thus, pure BiVO_4_ has certain inhibitory activity, and the antibacterial activity of PDA-rGO/BiVO_4_ nanocomposites is superior to that of pure BiVO_4_, which is consistent with previous studies [16,30]. The structure and morphology of the samples, including surface roughness, surface area, edge sharpness, sheet size, hydrophilicity, dispersion, and functionalization, may control the antibacterial properties of graphene-based nanocomposites [31,32,33]. Liao et al. [20] compared the inhibition rates of GO, PDA-RGO, and Ag-PDA-RGO on *E. coli*. The antibacterial activity of Ag-PDA-RGO nanocomposites was significantly higher than that of GO and PDA-RGO. The combination of Ag with PDA-rGO significantly improved the antibacterial activity of the nanocomposites. Therefore, the synergistic effect between the PDA-rGO and BiVO_4_ particles enhanced the antibacterial activity [16,30].

### 2.4. ROS Production

It is well-known that the existence of ROS can destroy the stable state of and cause oxidative damage to bacterial cells, ultimately leading to bacterial death [34]. Thus, a 2′,7′-dichlorofluorescein diacetate (DCFH-DA) fluorescence method was used to determine ROS production. Under light conditions, the normalized result of the ROS regeneration of individual bacteria treated with PDA-rGO/BiVO_4_ was significantly higher than those treated with BiVO_4_ and the control. When treated with BiVO_4_, it was only slightly higher than in the control. On the other hand, under dark conditions, the normalized result of the ROS regeneration of individual bacteria treated with BiVO_4_ was comparable to that of the control, but higher than that treated with PDA-rGO/BiVO_4_. When comparing light and dark conditions, the normalized result of the ROS regeneration of individual bacteria treated with PDA-rGO/BiVO_4_ was significantly higher under light conditions. When treated with BiVO_4_, and in the control group, ROS regeneration was significantly lower under light conditions than under dark conditions (Figure 4). Therefore, a greater generation of ROS is considered as the main reason for the enhanced antibacterial properties of the PDA-rGO/BiVO_4_ composite under light irradiation. This is in agreement with previous studies, which showed that modified BiVO_4_ can produce more ROS with high oxidation capacity, which is beneficial to the improvement of photocatalytic activity [35,36].

### 2.5. Microstructures of Bacteria

The morphological changes of *E. coli* before and after treatment with PDA-rGO/BiVO_4_ are shown in Figure 5. The survival *E. coli* displayed a short rod-like shape with a complete cell wall [37]. *E. coli* treated with PDA-rGO/BiVO_4_ exhibited an irregular ellipsoid shape with incomplete cell structure, indicating bacterial death [38]. Thus, oxidative damage to the cell body with increased ROS resulted in the destruction of the bacterial internal structure. The bacterial cell bodies showed obvious inhibition and adhesion to each other and, finally, the bacterial cell membrane lost its biological function. In conclusion, the ROS produced by PDA-rGO/BiVO_4_ under visible light irradiation did some damage to the bacteria. These results further confirm the analysis of antibacterial and ROS fluorescence detection.

### 2.6. Genes Expression

The expression levels of key genes in the PDA-rGO/BiVO_4_ treatment group and control group were compared under light conditions. *Thil*, *narU,* and *livK*, encode tRNA uridine 4-sulfurtransferase, nitrate/nitrite transporter, and L-leucine/L-phenylalanine ABC transporter periplasmic-binding protein, respectively, which are involved in the transmembrane transport of substances. Compared with the control group, the expression levels of *thil*, *narU*, and *livK* in the PDA-rGO/BiVO_4_ treatment group were down-regulated by 12.99, 13.33, and 3.58 times, respectively (Figure 6). *aceE* encodes the E1 subunit of pyruvate dehydrogenase, which catalyzes the formation of acetyl coenzyme A (CoA) from pyruvate and CoA with the participation of NAD^+^. *PflB* encodes pyruvate lyase, which catalyzes the reaction of acetyl CoA with formic acid to produce CoA and pyruvate. Compared with the control group, the expression level of *aceE* in the PDA-rGO/BiVO_4_ treatment group was up-regulated by 2.74 times, while the expression level of *pflB* was slightly decreased. The proteins encoded by *cyoC* and *cyoD* belonged to cytochrome C oxidase (complex IV) in the respiratory chain, and the proteins encoded by *atpC* belonged to ATP synthase (complex V). In the BiVO_4_ + rGO treatment group, the expression levels of *cyoC* and *cyoD* were up-regulated by 2.59 times, and the expression levels of *atpC* were down-regulated by 1.85 times.

Figure 7 exhibits the possible antibacterial mechanisms. The expression levels of *thil*, *narU*, and *livK* genes involved in the transmembrane transport were significantly down-regulated, which was consistent with a previous study [39]. PDA-rGO/BiVO_4_ can produce ROS after photocatalysis, which can lead to the destruction of the phospholipid components of the cell membrane by peroxidation, indicating that photocatalytic disinfection can destroy cell membranes, thus eliminating the transmembrane transport reactions on the cell membrane [40]. The expression level of *aceE* was significantly up-regulated, which was consistent with the results of Liang et al. [39], indicating the increased metabolic flux of CoA. Contrary to the results of Liang et al. [39], the expression level of *pflB* in the catalyzed CoA synthesis reaction was slightly reduced. However, changes in *aceE* and *pflB* can also reduce the content of CoA in cells. Matsunaga et al. found that, in the process of the TiO_2_-mediated photochemical sterilization of *E. coli* cells, the content of intracellular CoA decreased, indicating that the photooxidation of CoA is involved in the photocatalytic inactivation of bacteria [41]. Therefore, the photochemical sterilization process mediated by PDA-rGO/BiVO_4_ in this study also leads to cell death by reducing the content of CoA in cells. Complex III in the respiratory chain is the main production source of ROS. After PDA-rGO/BiVO_4_ photocatalytic treatment, cells produced a large amount of ROS. The up-regulation of cyoC and cyoD expression in complex IV might be in response to resisting ROS attacks in complex IV. Matsunaga et al. found that cell respiration was inhibited, oxidative phosphorylation was reduced, and ATP synthase was reduced during photocatalytic disinfection [41]. In this study, atpC expression level was down-regulated, confirming the inhibition of the respiratory chain.

## 3. Materials and Methods

### 3.1. Materials

The synthetic precursors used in this preparation were taken as analytical-grade chemicals and subsequently utilized without any further purification. The starting precursors were sodium metavanadate (NaVO_3_), bismuth nitrate pentahydrate (Bi(NO_3_)_3_·5H_2_O), ethylenediamine tetraacetic acid disodium salt (EDTA-2Na), HNO_3_, and double-distilled (DD) water, which were purchased from Shanghai Chemical Reagent Co., Ltd. (Shanghai, China). GO was obtained from Suzhou Carbon Graphene Technology Co., Ltd. (Suzhou, China). Dopamine hydrochloride was supplied by Bi De Pharmaceutical Technology Co., Ltd. (Shanghai, China). Tris-hydrochloric acid (Tris-HCl) was purchased from Beijing Jingke Hongda Biotechnology Co., Ltd. (Beijing, China).

### 3.2. Preparation of BiVO_4_

The fabrication of BiVO_4_ was prepared with reference to our previous work [42]. The synthetic precursor solution, with a pH of 7, contained Bi(NO_3_)_3_ (2.911 g), EDTA-2Na (2.500 g), and HNO_3_ (10 mL); then, the NaVO_3_ (0.732 g) solution was dropped under stirring conditions. The pH value was adjusted to 5 with 1M HNO_3_, then the whole solution was put into a micro reactor and reacted at 160 °C for 1 h under stirring to synthesize the BiVO_4_ (Figure 8). Finally, the product was washed with deionized water and dried at 25~30 °C overnight.

### 3.3. PDA-rGO/BiVO_4_ Nanocomposite Synthesis

PDA-rGO/BiVO_4_ was prepared according to a previously reported synthesis method [21]. Briefly, 50 mg DA, 100 mg GO and already prepared BiVO_4_ (100 mg) were added into 200 mL Tris-HCl solution (10 mM, pH = 8.5) and sonicated (150 W) for 15 min in an ice bath. The reaction solution was mechanically stirred for 24 h at 60 °C followed by 3× centrifugation (10,000× *g*) to remove excess polydopamine (PDA). Finally, the products were dried for 24 h at room temperature.

### 3.4. Characterization 

The size and structure of the samples was identified by a field emission scanning electron microscope (FE-SEM, ZEISS Gemini SEM 300, ZEISS, Oberkochen, Germany) within 8–10 kV accelerating voltage. EDX (Aztec X-Max50, Oxford Instruments, Oxford, UK) was used to confirm the elemental composition of the prepared nanocomposites. 

The FT-IR spectra of the nanoparticles was carried out on a TENSOR 27 FT-IR spectrometer (Bruker, Munich, Germany) within the range of 600–4000 cm^−1^ at a scanning resolution of 2 cm^−1^.

XRD (Bruker-AXS APEX2, Germany, analytical CuKαRigaku D/max 2550 PC, 40 kV, 300 mA) was used for studying the crystal structure and phase of the sample in the range of 5° to 85° with a 5° (2θ) per min scanning rate. 

The visible light photothermal response was acquired by utilizing an infrared thermometer (FLIR C3). Temperatures represent the direct output from the IR system. The temperature of the sample was recorded after being exposed to a 100 W xenon lamp (JJY-12610, Jujingyang, Shantou, China) for different time intervals: 0 min, 5 min, 10 min, 15 min, 20 min, and 25 min.

### 3.5. Antibacterial Ability Assays

*E. coli* MG1655 (ATCC700926) has been widely used and studied as a model bacterium and has clear genome information. *E. coli* MG1655 was used to assay the antibacterial ability of BiVO_4_ and PDA-rGO/BiVO_4_ in this work. *E. coli* was pre-incubated in 5 mL lysogeny broth medium (LB; Feiyu Bio, Nantong, China) at 37 °C and 200 rpm overnight. Then, 10 μL cultures (1 × 10^5^ colony-forming units per milliliter (CFU/mL)) were transferred into the 2 mL microfuge tube containing 400 μL medium. The final concentrations of BiVO_4_ and PDA-rGO/BiVO_4_ in the medium were adjusted to 100 mg/mL. LB medium without nano materials was used as the control. Under the condition of open cover, the microfuge tube was irradiated vertically with a xenon lamp for 20 min. Then, *E. coli* was cultured at 37 °C and 200 rpm for 12 h. To ensure the trigger efficiency of light on the nano materials to the bacteria, the same cultures were performed without light. All the tubes were covered with aluminized paper. Cell activity was measured using the MTT assay kit (Beyotime, Shanghai, China). Three independent assays were performed.

LB plate (agar, 15 g/L) was used to show the cell survival ability directly. The treated cells were adjusted to 10^3^ diluents. Then, 70 μL diluent was spread evenly on the LB plate and cultured overnight at 37 °C. Three independent assays were performed.

### 3.6. ROS Analysis

*E. coli* was cultured in 5 mL LB medium at 37 °C and 200 rpm overnight. One milliliter of the cultures was centrifuged at 4 °C and 8000× *g* for 3 min. The pellet was incubated with DCFH-DA, using the ROS Assay Kit (Beyotime, China). Then, 10 μL cells were added into 400 μL LB medium which contained 100 mg/mL BiVO_4_ or rGO/BiVO_4_. LB medium without materials was used as the control. The experiments were conducted both under exposure to a xenon lamp (20 min) and without light. Then, dichlorofluorescein (DCF) fluorescence distribution was detected by fluorospectrophotometer analysis at an excitation wavelength of 488 nm and at an emission wavelength of 525 nm (RF-5301PC, Shimadzu Co. Ltd., Kyoto, Japan). ROS content per MTT unit was calculated to show the ROS amount under the same cell level. Three independent assays were performed.

### 3.7. Detection of mRNA Levels of Key Genes

*E. coli* cells treated with PDA-rGO/BiVO_4_ and light were prepared according to “Antibacterial ability assays”. The initial cell concentration was 1 × 10^6^ CFU/mL and the PDA-rGO/BiVO_4_ concentration was 10 mg/mL. The *E. coli* cells cultured with only LB medium and light were used as controls.

One milliliter of the control cultures and two milliliters of the PDA-rGO/BiVO_4_ treated cultures were collected and centrifuged at 4 °C and 10,000× *g* for 6 min. After discarding the supernatant, the cells were washed twice with 5 mL sterile ddH_2_O. Total RNA was extracted using the TRIzol reagent (Ambion, Austin, TX, USA). The residual DNA was removed and the first-strand cDNA was synthesized in one pot using the Transcript All-in-One First-Strand DNA Synthesis SuperMix for qPCR (One-Step gDNA Removal) Kit (Tsingke, Beijing, China). Quantitative real-time PCR was performed using a set of 2 PCR primers with SYBR Green I Real-Time PCR (Solarbio, Beijing, China). The PCR analysis was carried out using a 7500 real-time PCR system (Applied Biosystems, Waltham, MA, USA). In all cases, *16S rRNA* was used as a reference gene, and the relative gene expression was calculated by the 2^−ΔΔCT^ method. The primers used in this experiment are indicated in Table 1. Five independent assays were performed.

### 3.8. Statistical Analysis

Statistical analysis of the experimental data was carried out using Origin 8.0 software (OriginLab, Northampton, MA, USA). Variance analysis or the *t*-test were used for comparison between groups. * Indicates a significant difference at *p* < 0.05. ** Indicates a highly significant difference at *p* < 0.01.

## 4. Conclusions

In summary, PDA-rGO/BiVO_4_ hybrid materials were successfully prepared by a hydrothermal self-polymerization reduction method and BiVO_4_ particles were evenly distributed on the rGO surface. The combination of PDA-rGO and BiVO_4_ effectively improved the utilization of the visible-light-driven thermal effect. The photocatalytic inactivation of *E. coli* with the synthetic PDA-rGO/BiVO_4_ was superior to that of pure BiVO_4_ under visible light irradiation. The DCFH-DA fluorescence method was used to determine the production of ROS. Under light conditions, PDA-rGO/BiVO_4_ produced more ROS with high oxidation capacity, which was beneficial to the improvement of photocatalytic activity. According to the morphology changes in the SEM images of *E. coli* MG1655, *E. coli* treated with PDA-rGO/BiVO_4_ exhibited an irregular ellipsoid shape with an incomplete cell structure, indicating bacterial death. Mechanistically, the ROS produced by the PDA-rGO/BiVO_4_ penetrated the bacterial cell membrane and caused oxidative damage, blocked the transmembrane transport of bacteria, and inhibited the respiratory chain, which was confirmed by ROS fluorescence detection, morphological changes, and gene expression.

## Figures and Tables

**Figure 1 ijms-23-07712-f001:**
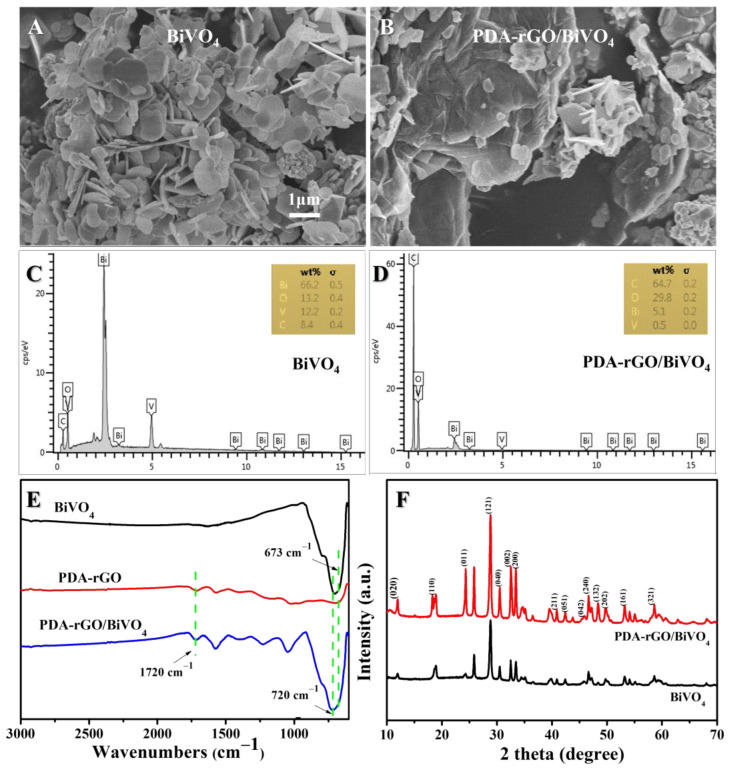
SEM images of (**A**) BiVO_4_, (**B**) PDA-rGO/BiVO_4_; EDX analysis of (**C**) BiVO_4_, (**D**) PDA-rGO/BiVO_4_; (**E**) FT-IR spectra of BiVO_4_, PDA-rGO and PDA-rGO/BiVO_4_; and (**F**) XRD patterns of BiVO_4_ and PDA-rGO/BiVO_4_.

**Figure 2 ijms-23-07712-f002:**
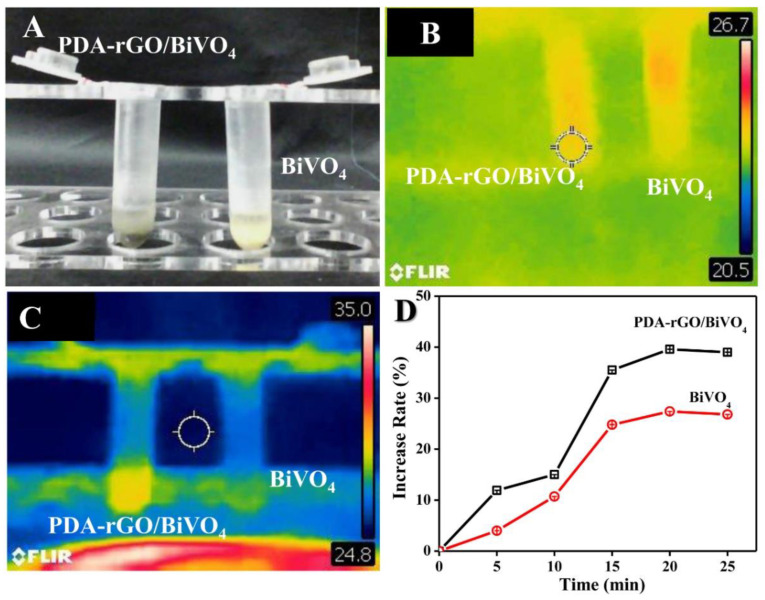
(**A**) Photograph and (**B**) infrared thermal images of the BiVO_4_ and PDA-rGO/BiVO_4_ composite dispersed in double-distilled water before xenon lamp exposure. (**C**) Infrared thermal images of the rGO/BiVO_4_ composite after xenon lamp exposure for 10 min. (**D**) Measured temperature of the composite calculated from infrared thermal images. The symbols are increase rates of BiVO_4_ and PDA-rGO/BiVO_4_.

**Figure 3 ijms-23-07712-f003:**
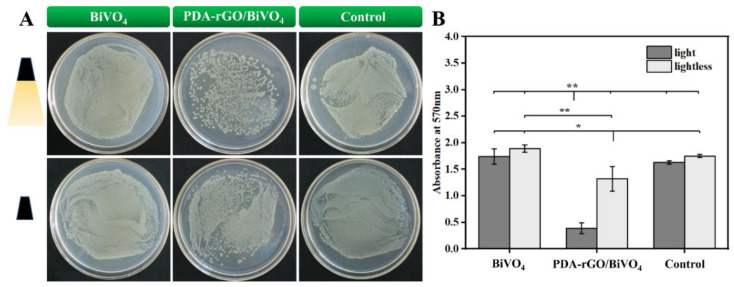
The antibacterial effect of PDA-rGO/BiVO_4_ and BiVO_4_ with or without xenon light irradiation was determined by flat colony counting (**A**) and MTT assay (**B**) (* *p* < 0.05 and ** *p* < 0.01 indicates significant difference).

**Figure 4 ijms-23-07712-f004:**
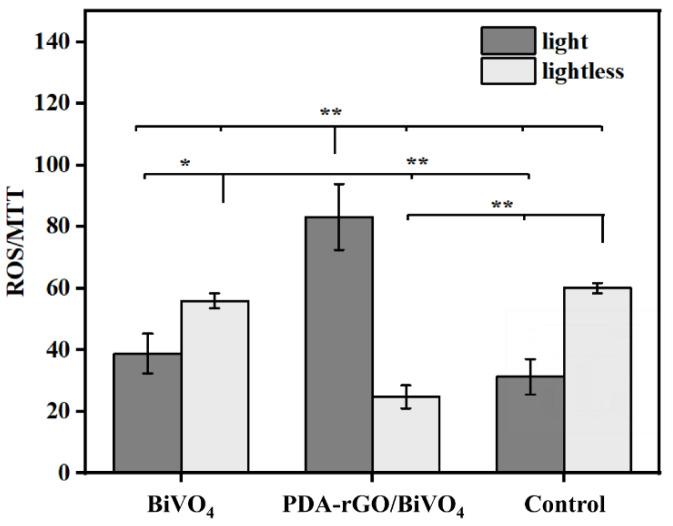
ROS activity normalized to bacteria numbers (* *p* < 0.05 and ** *p* < 0.01 indicate significant difference).

**Figure 5 ijms-23-07712-f005:**
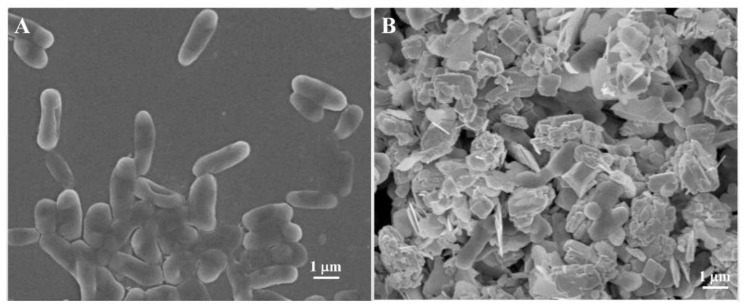
SEM images of *E. coli* (**A**) and *E. coli* treated with the PDA-rGO/BiVO_4_ composite (**B**).

**Figure 6 ijms-23-07712-f006:**
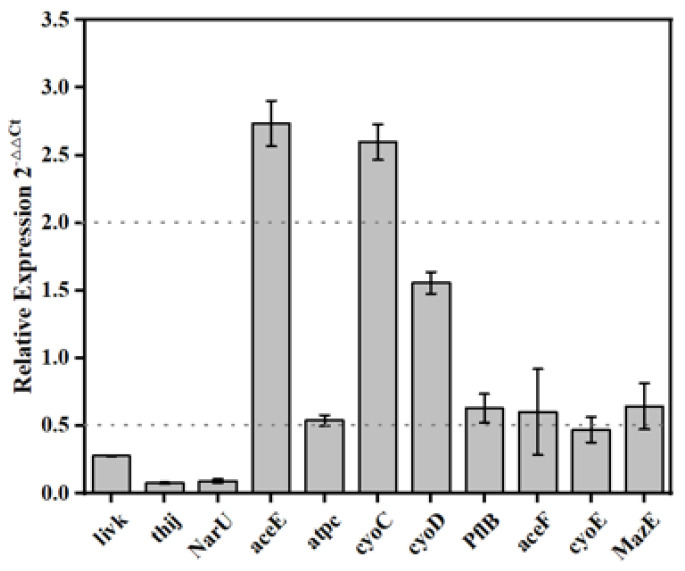
Gene expression of *E. coli* treated with the PDA-rGO/BiVO_4_ composite.

**Figure 7 ijms-23-07712-f007:**
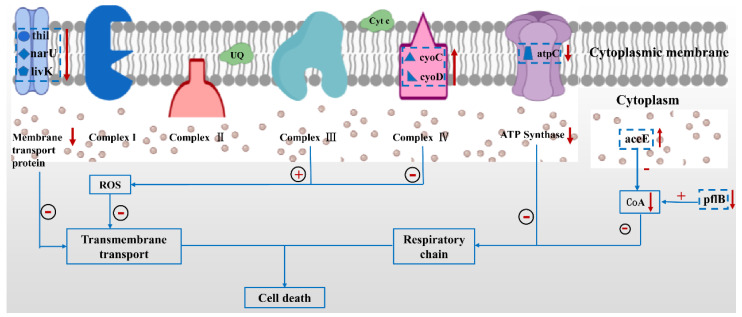
Schematic of the proposed antibacterial mechanism of PDA-rGO/BiVO_4_ against *E. coli.* The up/down arrow indicates an increase/decrease in expression, the plus/minus sign indicates a positive/negative effect.

**Figure 8 ijms-23-07712-f008:**
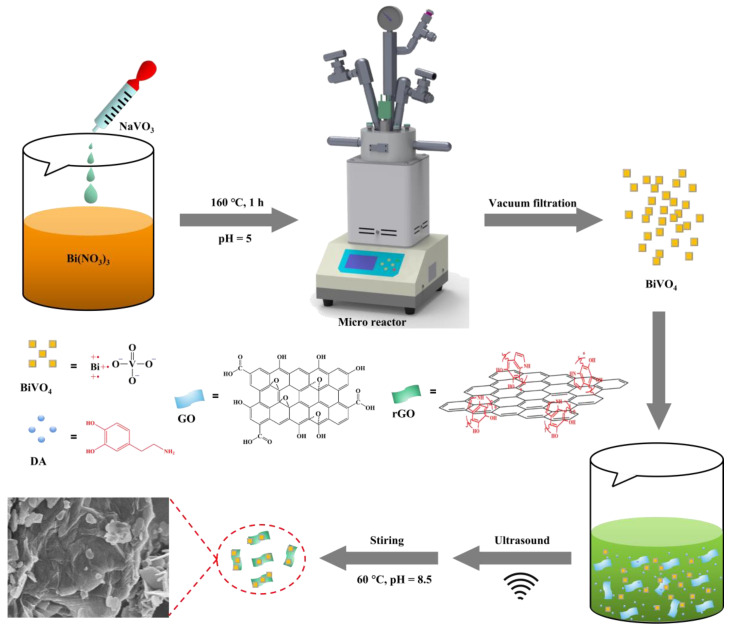
Preparation of PDA-rGO/BiVO_4_ nanocomposites.

**Table 1 ijms-23-07712-t001:** Primers used in qRT-PCR.

Gene Name	Description	Forward (5′→3′)	Reversed (5′→3′)
*thil*	tRNA uridine 4-sulfurtransferase	GGCTGTACGTGCTATAAAAG	GCGCAAAAGATGAAAACCAG
*narU*	nitrate/nitrite transporter	GACATAAACGGAAAGAGCCG	AAAAGCCAAACAAGGGAGCG
*aceE*	pyruvate dehydrogenase E1 component	CGTCAAGGGCTGGCATAGAT	GGTCGTCTGACTCAGGAGCA
*pflB*	pyruvate formate-lyase	ATGCTCCCACTGCTCAGGAA	GCTTAATGAAAAGTTAGCCACA
*cyoD*	cytochrome bo3 ubiquinol oxidase subunit 4	GCAAACAGAAGTGGCACGAT	CTGGCAATGGCAGTGGTACA
*cyoC*	cytochrome bo3 ubiquinol oxidase subunit 3	CTACCACGTTTAGCGGGCAG	GGCTTCCTGTCAGCGTTCTT
*atpC*	ATP synthase F1 complex subunit epsilon	GTACCTTCGCTTTGCATTCC	TATCTATCTGTCTGGCGGCATT
*livK*	L-leucine/L-phenylalanine ABC transporter periplasmic binding protein	GTAGGTGCCGTCGGTTCACT	CGTGATTGGGCCGCTGAACT
*16S rRNA* (reference)	16S ribosomal RNA	CCAACACGGGAACTCCGCAC	CTGGACGAAGACTGACGCTC

## Data Availability

All relevant data are presented in the manuscript; raw data are available upon request from the corresponding author.

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
