# Peer review of "Visible-Light-Driven Antimicrobial Activity and Mechanism of Polydopamine-Reduced Graphene Oxide/BiVO4 Composite"

_ijms, 2022, doi:10.3390/ijms23147712_

Round 1
Reviewer 1 Report
Dear Authors,
The paper deals with a photocatalytic antibacterial composite of polydopamine reduced graphene 9 oxide (PDA-rGO)/BiVO4. The morphology and physicochemical properties were characterized by SEM, EDS, FT-IR, XRD and Escherichia coli (E. coli) MG1655 was selected as the model bacteria. Its antibacterial performance was tested by flat colony counting and MTT method under light irradiation.
The photocatalytic inactivation of E. coli with the synthetic PDA-rGO/BiVO4 was superior to that of pure BiVO4 under visible-light irradiation. Meanwhile, the ROS produced by the PDA- rGO/BiVO4 penetrated the bacterial cell membrane and damaged it. ROS fluorescence detection and the morphology changes confirmed the blocking transmembrane transport of bacteria and inhibited the respiratory chain.
The paper topic is of utmost importance considering the current pandemic context and humanity healthy.
My comments are as following
- first of all in any study you want to be published, before using an abbreviation it must be defined
- the introduction section should be extended by references related also to other similar materials with antimicrobial activity
- the manuscript text is required to be reorganized
- the materials, the characterization techniques should be as separate subsections but before the chapter of results and discussion
- with respect to all materials and to all characterization instruments and equipments used there is required to be provided all information of them (for instance related to materials the purity, th company name, the country provenience and so on)
- lines 69 - 70 you said:
The diffraction pattern of BiVO4 and PDA-rGO/BiVO4 nanocomposite clearly shows 69 sharp peaks of tetragonal zircon phase
Where from the zircon phase?
Line 207, at 3.2 Preparation of BiVO4, you wrote:
Finally, the product was washed with deionized water and dried.
But how it was dried?
You should add this information at this place.
Conclusions section is too short, please reformulate it
For instance, you said
The ROS produced by the PDA- rGO/BiVO4 penetrated the bacterial cell membrane and caused oxidative damage, blocked transmembrane transport of bacteria and inhibited the respiratory chain, which were confirmed by ROS fluorescence detection, the morphology changes and gene expression.
There is the need to detail and outline the results of ROS fluorescence detection.
More precisely which were the morphology changes supporting the oxidative damage of the bacterial membrane cell?
Author Response
Response to Reviewer 1's Comments
Dear Authors,
The paper deals with a photocatalytic antibacterial composite of polydopamine reduced graphene oxide (PDA-rGO)/BiVO4. The morphology and physicochemical properties were characterized by SEM, EDS, FT-IR, XRD and Escherichia coli (E. coli) MG1655 was selected as the model bacteria. Its antibacterial performance was tested by flat colony counting and MTT method under light irradiation.
The photocatalytic inactivation of E. coli with the synthetic PDA-rGO/BiVO4 was superior to that of pure BiVO4 under visible-light irradiation. Meanwhile, the ROS produced by the PDA- rGO/BiVO4 penetrated the bacterial cell membrane and damaged it. ROS fluorescence detection and the morphology changes confirmed the blocking transmembrane transport of bacteria and inhibited the respiratory chain.
The paper topic is of utmost importance considering the current pandemic context and humanity healthy.
My comments are as following
- first of all in any study you want to be published, before using an abbreviation it must be defined
Response: Thank you for your comments. It was corrected. Abbreviations were defined when they first appeared.
- the introduction section should be extended by references related also to other similar materials with antimicrobial activity
Response: Thank you for your comments. It was revised according to the suggestion.
- the manuscript text is required to be reorganized
Response: Thank you for your comments. It was corrected.
- the materials, the characterization techniques should be as separate subsections but before the chapter of results and discussion
Response: Thank you for your comments. The required order of the journal of IJSM is introduction, results and discussion, materials and methods and conclusions. So we put the method behind the results and discussion according to the requirements of the journal.
- with respect to all materials and to all characterization instruments and equipments used there is required to be provided all information of them (for instance related to materials the purity, th company name, the country provenience and so on)
Response: Thank you for your comments. It was corrected in 3.1. Materials. Other marks were in the corresponding part.
- lines 69 - 70 you said:
The diffraction pattern of BiVO4 and PDA-rGO/BiVO4 nanocomposite clearly shows sharp peaks of tetragonal zircon phase
Where from the zircon phase?
Response: Thank you for your comments. The presence of tetragonal zircon phase BiVO4 in nanocomposite is confirmed by the (200) plane at 24.6° and the diffraction pattern is in accordance with the JCPDS card number (14-0133) of BiVO4. This part is also re-written in the manuscript.
Line 207, at 3.2 Preparation of BiVO4, you wrote:
Finally, the product was washed with deionized water and dried.
But how it was dried?
You should add this information at this place.
Response: Thank you for your comments. It was corrected (Line 227).
Conclusions section is too short, please reformulate it
Response: Thank you for your comments. It was corrected.
For instance, you said
The ROS produced by the PDA- rGO/BiVO4 penetrated the bacterial cell membrane and caused oxidative damage, blocked transmembrane transport of bacteria and inhibited the respiratory chain, which were confirmed by ROS fluorescence detection, the morphology changes and gene expression.
There is the need to detail and outline the results of ROS fluorescence detection.
More precisely which were the morphology changes supporting the oxidative damage of the bacterial membrane cell?
Response: Thank you for your comments. It was corrected in conclusions.
Reviewer 2 Report
Add more data to the abstract.
Use better keywords.
Compare the results with similar studies.
Update references.
It is recommended to evaluate PDA-rGo properties without BiVo4 and compare the results to understand the PDA-rGO effect.
Author Response
Reviewer 2
Add more data to the abstract.
Use better keywords.
Compare the results with similar studies.
Update references.
Response: Thank you for your comments. It was corrected.
It is recommended to evaluate PDA-rGO properties without BiVO4 and compare the results to understand the PDA-rGO effect.
Response: Thank you for your comments. PDA-rGo had no visible light response. Our focus in this paper was to prepare visible light responsive nanocomposites. Combination of BiVO4 with PDA-rGO was confirmed by hydrothermal and self-polymerization reduction method, which can improve the photothermal performance under visible-light irradiation. Liao et al. reduced GO with dopamine (DA), and then synthesized Ag-PDA-RGO nanocomposites in situ by adding AgNO3. The inhibition rates of GO, PDA-RGO and Ag-PDA-RGO on E. coli were compared. Compared with GO and PDA-RGO, the antibacterial activity of Ag-PDA-RGO nanocomposites was significantly improved, and the antibacterial rate of E. coli reached 90.9%. Combination of Ag with PDA-rGO was significantly improved the antibacterial activity of nanocomposites. PDA-rGO also combined with BiVO4 to inhibit the recombination of electron-hole pairs and improve photocatalytic antibacterial activity by a synergetic effect.
Liao J., He S., Guo S., Luan P., Mo L., Li J. Antibacterial performance of a mussel-inspired polydopamine-treated Ag/graphene nanocomposite material. Materials (Basel) 2019, 12, 3360.
Reviewer 3 Report
Overall the study is well-conceived and the manuscript well-written. However, it requires some minor editing before it can be published. My thoughts are as follows:
- please check references, e.g. position 22 is only in the Materials and Methods and is not mentioned earlier,
- page 8, line 203 - us work. [22]The – please correct - add a space and a period in the appropriate place
- please provide suppliers of reagents like HNO3 (page 8, line 203), Tris-HCl (page 9, line 215),
- please provide a supplier of E. coli bacteria,
- please provide a supplier of lysogeny broth medium,
- what are the intensity and parameters of the xenon lamp?
- page 10, line 259: ( RF-5301PC, Shimadzu Co. LTD, Japan) – please remove space between the RF-5301PC).
Author Response
Reviewer3
Overall the study is well-conceived and the manuscript well-written. However, it requires some minor editing before it can be published. My thoughts are as follows:
- please check references, e.g. position 22 is only in the Materials and Methods and is not mentioned earlier,
Response: Thank you for your comments. It was corrected. We reordered the references.
- page 8, line 203 - us work. [22]The – please correct - add a space and a period in the appropriate place
Response: Thank you for your comments. It was corrected (page 8, line 222).
- please provide suppliers of reagents like HNO3 (page 8, line 203), Tris-HCl (page 9, line 215),
Response: Thank you for your comments. It was corrected in 3.1. Materials (page 8, line 215, 219).
- please provide a supplier of E. coli bacteria,
Response: Thank you for your comments. It was corrected in Materials and Methods 3.5 (page 10, line 252).
- please provide a supplier of lysogeny broth medium,
Response: Thank you for your comments. It was corrected in Materials and Methods 3.5 (page 10, line 256).
- what are the intensity and parameters of the xenon lamp?
Response: Thank you for your comments. It was corrected in Materials and Methods 3.4 (page 9, line 250-251).
- page 10, line 259: (RF-5301PC, Shimadzu Co. LTD, Japan) – please remove space between the RF-5301PC).
Response: Thank you for your comments. It was corrected (page 10, line 277).
Round 2
Reviewer 2 Report
The authors have addressed all comments and I recommend publishing the article.